

# Investigation of role of antisite disorder in pristine cage compound FeGa$_3$

C. Kaufmann Ribeiro [1], L. Mello [1], V. Martelli[1], D. Cornejo [2], M. B. Silva Neto [3], E. Fogh [4], H. M. Rønnow [4] and J. Larrea Jiménez [1⋆]

**1** Laboratory for Quantum Matter under Extreme Conditions, Institute of Physics, University of São Paulo, São Paulo, Brazil
**2** Institute of Physics, University of São Paulo, São Paulo, SP, Brazil
**3** Instituto de Fisica, Universidade Federal do Rio de Janeiro , Rio de Janeiro, Brazil
**4** Laboratory for Quantum Magnetism, Institute of Physics, Ecole Polytechnique Fédérale de Lausanne (EPFL), Lausanne, Switzerland

⋆ larrea@if.usp.br

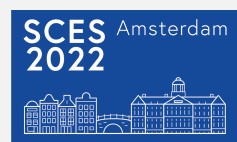

## Abstract

The role of controlled Fe antisite disorder in the narrow gap semiconductor FeGa$_3$ has been investigated. Polycrystalline samples were synthesized by the combination of arc-melting furnace and successive annealing processes. Deviations from occupation numbers of Fe and Ga sites expected in the pristine compound were obtained from X-ray data using Rietveld refinement analysis. Besides that, electrical transport and magnetization measurements reveal that hierarchy in Fe and Ga site disorder tunes the ground state of FeGa$_3$ from paramagnetic semiconductor to a magnetic metal. These findings are discussed inside the framework of Anderson localization in the vicinity of metal-semiconductor transitions and spin fluctuations.

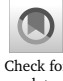

# 1 Introduction

Atomic disorder is an ubiquitous feature in realistic quantum materials. The presence of random defects, like vacancies, dislocations and impurities, induced during sample preparation are usually seen as an unfavorable mechanisms to stabilize novel quantum states driven by quantum phase transitions [1]. However, a controlled modification of disorder can be used as tuning control parameter to investigate emergent quantum phenomena that go beyond the scope of decoherence and localization of fermions inducing Anderson-localized states [2]. This is the case, for instance, of antisite disorder in double pervoskites and low-dimensional oxides compounds, where exotic magnetic and electronic states emerge [3,4]. Last but not least, the investigation of the influence of antisite disorder on the correlation between electron and spin degrees of freedom is timely to establish alternative routes to uncover new quantum states of matter from a perspective of non-canonical quantum phase diagrams.

Considering the existing variety of synthetic crystal symmetries, compounds with cage-like structure offer fashionable platforms to investigate diverse problems in quantum materials, such as, collective spin and orbital states close to a quantum critical point (QCP) [5], anomalous electron-electron scattering process [6] and the interplay between magnetic frustration and disorder to stabilize spin glass phases [7].

Among cage-compounds, $FeGa_3$ is a strongly correlated narrow-gap semiconductor candidate and has been widely studied because of its unconventional magnetic and electronic properties that leads as a promising thermoelectric material [8–12]. Electrical resistivity measurements show an intrinsic activation gap $\approx 0.4$ eV and indicate the presence of in-gap donor states just below the conduction band in pristine single-crystals [13, 14]. Although $FeGa_3$ has been widely accepted as a semiconductor, the inclusion of subtle amount of disorder might drive $FeGa_3$ to $p$-type metal [15]. Magnetic susceptibility and Mössbauer spectroscopy confirms the absence of long range magnetic interaction in pristine $FeGa_3$ [14,16], in agreement with Density-Functional Theory (DFT) calculation [17]; while, neutron scattering shows signatures of a magnetic ground state, interpreted as a complex antiferromagnetic structure [18]. In addition, the chemical doping substitution of Ge atoms in Ga sites in the solid solution $FeGa_{3-x}Ge_x$ induces a transition from a diamagnetic semiconductor to a ferromagnetic metal, with a putative QCP at $x_c \approx 0.14$. The coexistence between long-range and short-range magnetic order above $x_c$ suggests that disorder induced by random Ge substitution plays an important role in the zero-temperature phase transition and its concomitant magnetic ground states [19]. More recently, first-principle electronic band calculations propose that a complex structure of metallic in-gap states driven by subtle Fe disorder might be responsible for the origin of very tiny magnetic moments likely interacting as a ferromagnetic ground state [20,21].

Here, we present a study of the influence of Fe antisite disorder in the electronic and magnetic physical properties of $FeGa_3$ polycrystals. Our results show that the amount of Fe antisite disorder can be controlled optimizing the annealing process, which turns out in the inclusion of tiny excess of Fe atoms in one of the non-equivalent Ga atomic sites.

# 2 Experimental

Polycrystalline samples with Fe antisite disorder named as $Fe_{1+\delta}Ga_3$ (with nominal $\delta = 0.04$) was prepared by solid-state reaction using an arc-melt furnace with a successive annealing process. High purity Ga and Fe pallets were melted in a cold water-cooled copper crucible under an argon atmosphere at ambient pressure. The as-cast samples were ground in an agate mortar, and the powder subsequently homogenized in size. For the annealing, the samples were encapsulated in a quartz tube under vacuum $\approx 10^{-5}$ mTorr, treated at different annealing

temperature $T_a$ for 24h, and cooled down to ambient temperature at a rate of 0.4 °C/min.

XRD diffraction were collected with a Diffractometer Rigaku Ultima III with angular step of 0.02° and $K_\alpha$-Cu wavelength. Energy dispersive x-ray spectroscopy (EDS) confirms the expected $\delta$ off-stoichiometric revealed by our Rietveld refinement analysis of XRD data within 5 % of uncertainty. Magnetization is measured at ambient temperature using a vibrating-sample-magnetometer (VSM) and DC magnetic field up to 1.5 T. For electrical transport measurements samples were compressed in circular pellets of 3 mm in diameter. Electrical resistivity ($\rho$) is measured in Van-der-Pauw configuration down to 16 K.

## 3   Results

Figure 1 shows the XRD patterns of the powdered samples at different annealing temperatures, $T_a$. The P4$_2$/mnm space group symmetry of the FeGa$_3$ phase is confirmed by Rietveld refinement (RR) for all $T_a$. No spurious phase is observed in the resolution of our XRD instrumentation. The lattice parameters in the as-cast sample $a = 6.2645(4)$ Å and $c$=6.5568(7) Å are in agreement with those values reported in literature for single-crystals [21]. Under annealing we did not observed significant change in the lattice parameter.

Our starting approach considers that the $P4_2/mnm$ space group symmetry arranges Fe and Ga atoms at three inequivalent Wyckoff sites $4f$, $8j$ and $4c$ [18]. In the case of the pristine compound FeGa$_3$, the disorder is not taken into account, resulting in $4f$ sites to be fully occupied by Fe atoms, whereas $8j$ and $4c$ sites host two inequivalent Ga atoms. On the other hand, when antisite disorder is considered, there are two possible arrangements: either Fe atoms might also occupy the $8j$ and $4c$ sites or the two inequivalent Ga sites extent their occupation at the $4f$ sites. If the former scenario sets in then Fe antisite disorder is formed and accounts for the tiny excess of Fe content in proportion to the pristine compound FeGa$_3$.

In order to evaluate the presence of antisite disorder, we compute the crystal structure of FeGa$_3$ using a careful RR analysis of our XRD data. Special attention is paid to the site occupancy number (SON) obtained in our RR results. Because SON is defined as the chemical occupancy times site multiplicity and normalized to the multiplicity of the general position, then the occupancy number and its influence in the quality of the RR-XRD patterns dictate the most appropriate atomic arrangement inside the primitive cell. By comparison of all the three different input-RR patterns, i.e; non-disordered pristine compound, Ga antisite disorder and Fe antisite disorder, the latter gives a most accurate description of our experimental data for all samples at different conditions of synthesis (see Fig. 1). Besides, due to our good quality XRD data, we used the Bragg R-factor ($|R_B|$) as another indicator to validate our initial proposed model including Fe antisite disorder. Because $|R_B|$ compares the observed and calculated integrated intensity (the latter also connected with the structural factor) [22], we found the lowest $|R_B|$ values (see table 1) when Fe antisite disorder is included at the Wickoff position $8j$ and $4c$.

Fig. 2 depicts the variation of the occupancy number as function of the annealing temperature and compares with as-cast sample. The reduction of the SON with the annealing temperature up to $T_a \approx 600$ °C indicates that Fe antisite disorder reaches a stable quantity nearly above this temperature. It is also remarkable that Fe antisite disorder prefers to displace Ga atoms mainly at the $8j$ sites.

Another evidence of the presence of Fe antisite disorder can be inferred from the computation of the $q$-vector dependence of the structure factor $S(q)$ [23]:

$$S(Q) = \sum_{n=1}^{n} O_{Ga}^n f_{Ga}(Q) e^{-2\pi i(h\mu_n + k\nu_n + lw_n)} + O_{Fe}^n f_{Fe}(Q) e^{-2\pi i(h\mu_n + k\nu_n + lw_j)}. \tag{1}$$
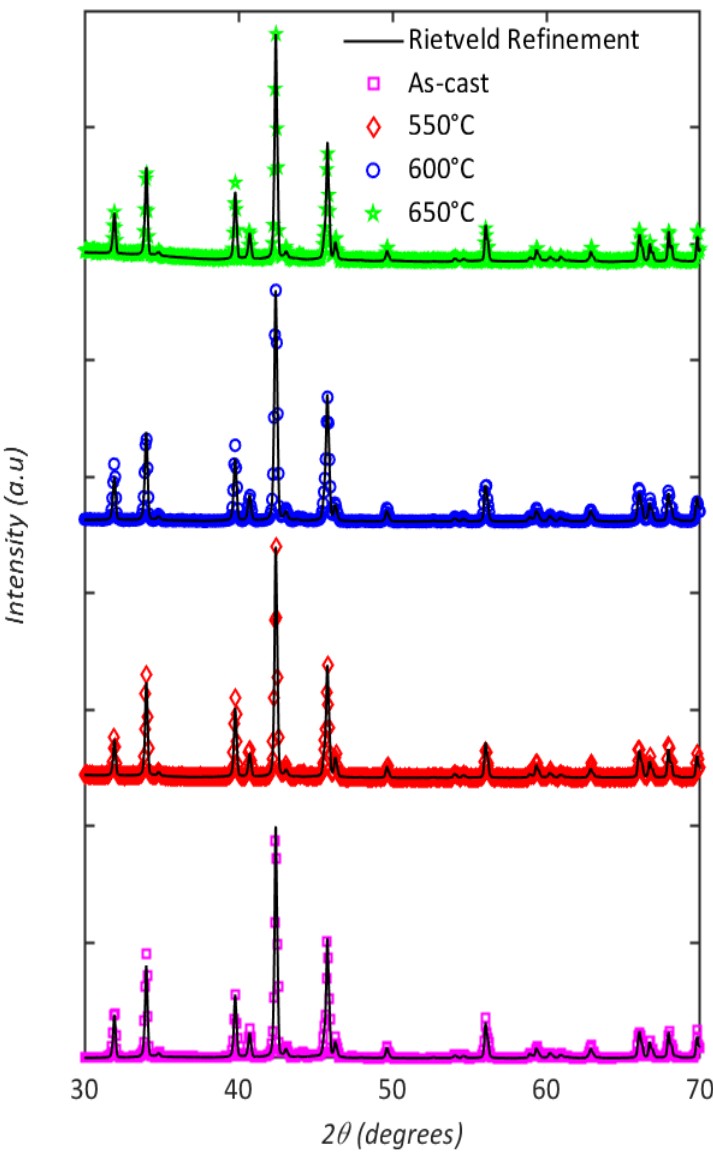

Figure 1: Experimental (open points) and calculated (black line) XRD patterns of as-cast and annealed FeGa$_3$ polycrystals.

Where $O_{Ga,Fe}$ and $f_{Ga,Fe}(Q)$ represent the atomic occupancy and the atomic form factor of Ga and Fe atoms, respectively. The terms in the imaginary exponential are the fractional position of nth-atomic site in the primitive unit cell $(u_n, v_n, w_n)$ and the reflection plane index $(h, k, l)$. The summation extends over all different atomic species at different sites in the primitive lattice. Given the Gaussian interpolation methods [24] we inferred the $f_{Ga}(Q)$ and $f_{Fe}(Q)$ values. The other parameters are quoted from symmetry operations of the $P4_2/mnm$ space group for the inequivalent sites $8j$, $4c$ and $4f$ as it is the case of $(u_n, v_n, w_n)$, as well as, from analogue equivalence between occupancy number in our RR analysis and those atomic occupancy to be used in eq. 1. Starting from the pristine compound FeGa$_3$, we use those values expected for single crystal [18]; i.e, $O_{Ga}^{8j} = O_{Ga}^{4c} = 1$ for Ga atoms at site $8j$ and $4c$ while $O_{Fe}^{4f} = 1$ for Fe atoms at site $4f$. Thus $O_{Ga} = O_{Fe} = 0$ if Ga and Fe atoms occupy different Wyckoff sites. In contrast when Fe antisite disorder takes place, $O_{Fe}^{4f} = 1$ and $0 \leq O_{Fe}^{4c}, O_{Fe}^{8j} \leq 1$

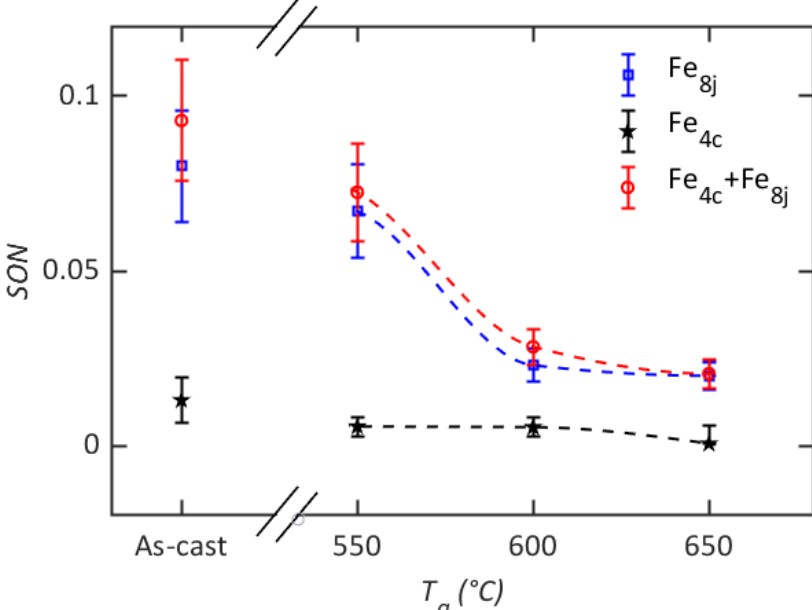

Figure 2: Site occupation number (SON) of Fe antisite as function of annealing temperature $T_a$ and compared with SON for as-cast sample. Fe atoms can occupy different non-equivalent crystallographic sites. The dashed lines are guide for the eyes.

constraining the Ga occupancy $O_{Ga}^n = 1 - O_{Fe}^n$ for any Wyckoff site. After a proper comparative normalization of SON values (see fig. 2), obtained from our RR analysis we include the atomic occupancy of Ga and Fe and determine the structure factor at different annealing conditions. $S(Q)$ for pristine $FeGa_3$ in the presence of Fe antisite disorder for as-cast and annealed samples are depicted in figure 3. As we can observe in fig. 3, oscillatory deviations from that $S(Q)$ expected for the pristine compound become less pronounced with the increasing of annealing temperature, at least up to 650 °C. Together with the variation of the occupancy number (see fig. 2) both SON and S(Q) feature that the Fe antisite disorder is controlled by annealing temperature until Fe atoms displace completely the Ga counterpart at $8j$ site.

Table 1: Bragg R-factor ($|R_B|$) from RR with and without inclusion of Fe antisite disorder.

|  | As-cast | 550 °C | 600 °C | 650 °C |
|---|---|---|---|---|
| Without Fe antisite | 7.36% | 10.23% | 9.89% | 13.37% |
| With Fe antiste | 6.92% | 9.7% | 9.57% | 13.27% |

On the other hand, the linear field dependence of the magnetization ($M$) per unit cell volume ($V_{unit\ cell}$), shown in 4, reveals a paramagnetic behavior in the presence of antisite disorder at room temperature (295 K). The increase of $T_a$ reduces the magnetic susceptibility $\frac{dM}{d\mu_0 H}$ in the samples, which might indicate a reduction of diluted magnetic moments if they likely formed under the influence of Fe antisite disorder.

Other features about the role of Fe antisite disorder can be inferred from the temperature-dependence of the electrical transport $\rho(T)$ shown in figure 5. Starting from the as-cast specimen, $\rho(T)$ shows a metallic behavior. The electrical resistivity (see figure 5) shows that the annealing induces significant changes in the electronic states mainly below 200 K. Successive annealing induces an upturn on $\rho(T)$ at $T \approx 200K$, which is attributed to a semiconducting extrinsic response due to in-gap donor states. The metal-semiconductor transition is likely

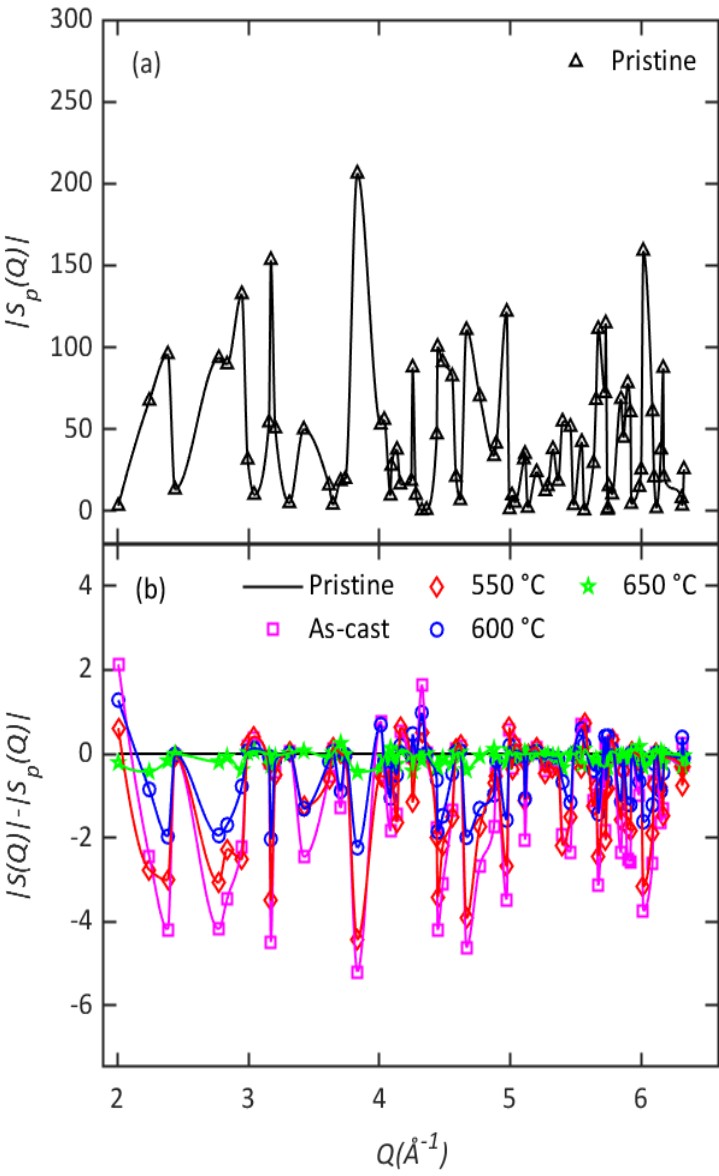

Figure 3: (a) Structural factor for the pristine FeGa$_3$ sample (b) Difference between the structural factor $S(q)$ for FeGa$_3$ polycrystalline samples annealed with different temperatures with respect to the structure factor expected for the pristine FeGa$_3$ polycrystal, $S_p(q)$. The continuous black line evaluates $S_p(q) - S_p(q)$.

related to the fact that the annealing decreases the concentration of Fe antisite. The Arrhenius law can well describe the electrical resistivity of the annealed samples, $\rho \propto e^{E_g/2k_bT}$, in the range of 100 K-160 K (see figure 5 (b)), where $E_g$ is the energy separation from in-gap donor states to the conduction band. On the contrary for the as-cast sample, the metallic behavior allows to describe $\rho(T) \propto A_f T^2$ (figure 5 (c)) resulting with a quadratic coefficient $A_f \approx 0.38(2)\ \mu\Omega cm/K^2$.

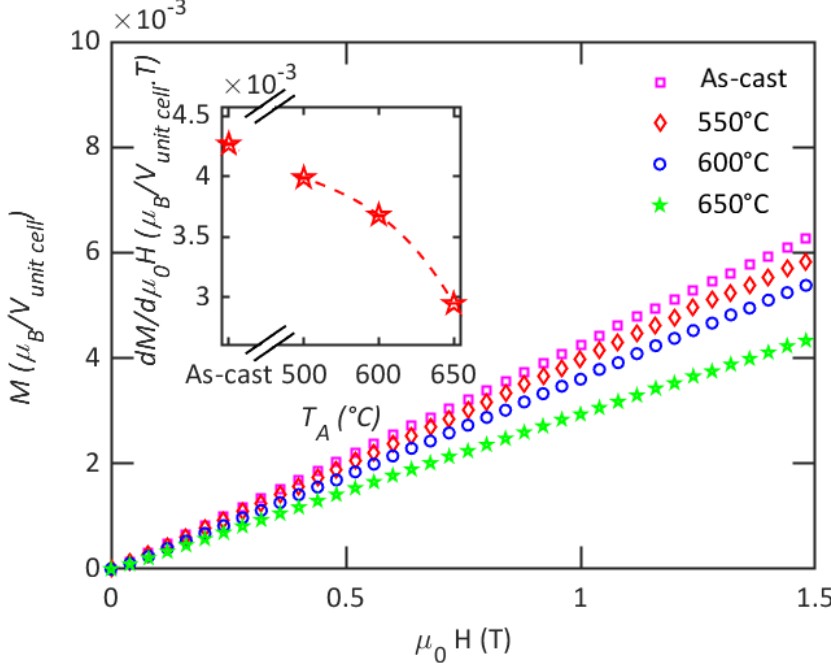

Figure 4: Field dependence of the magnetization per unit cell volume given at room temperature (295 K) for different annealing temperatures. The inset shows the annealing temperature-dependence of the magnetic susceptibility at ambient temperature.

## 4 Discussion

Recent Density Functional Theory (DFT) calculations have reported the influence of intrinsic disorder in $FeGa_3$ on the formation of in-gap states and diluted magnetic moments [20]. In addition, low-temperature magnetization measurements have shown the development of ferromagnetic ordering for non-stoichiometric Fe-rich $FeGa_3$ single crystals where the formation of this magnetic state was addressed to disorder-induced in-gap metallic states [21]. Previous electrical transport and spin-lattice relaxation experiments have indicated the presence of magnetic in-gap states intrinsic to $FeGa_3$ [25]. However, in all these previous results where disorder influences the pristine ground state in $FeGa_3$, there is an absence of investigation if disorder arises from antisite or ramdomic configuration of atoms.

   In the present study XRD, indicates that the density of antisite Fe defects decreases with annealing temperature, and induces a reduction of in-gap states near the Fermi energy, as well as the decrease of the magnetic susceptibility at ambient temperature.

   The scenario of in-gap states can also be used to explain the metal-semiconductor transition observed in our investigation. The observed $\rho(T)$ with $E_g$ at some decades of meV might indicate a type of metal-semiconductor Anderson transition, which occurs when the number of defects on the sample increases to a critical concentration called the percolation threshold. Below the critical concentration of defects, the in-gap states are localized due to the Anderson-localization effect [2]. On the other hand, as discussed by Mott, as the number of impurities approaches the critical concentration, the mobility edges shift towards the band edges and above a critical point, all the impurity states become delocalized, inducing a metallic behavior in the system [26]. Therefore, as the annealing temperature increases and the number of Fe-based defects are reduced, the band-edges of the impurity band (IB) shrink, separating the localized in-gap states from the delocalized extended states driving IB within the semiconduc-

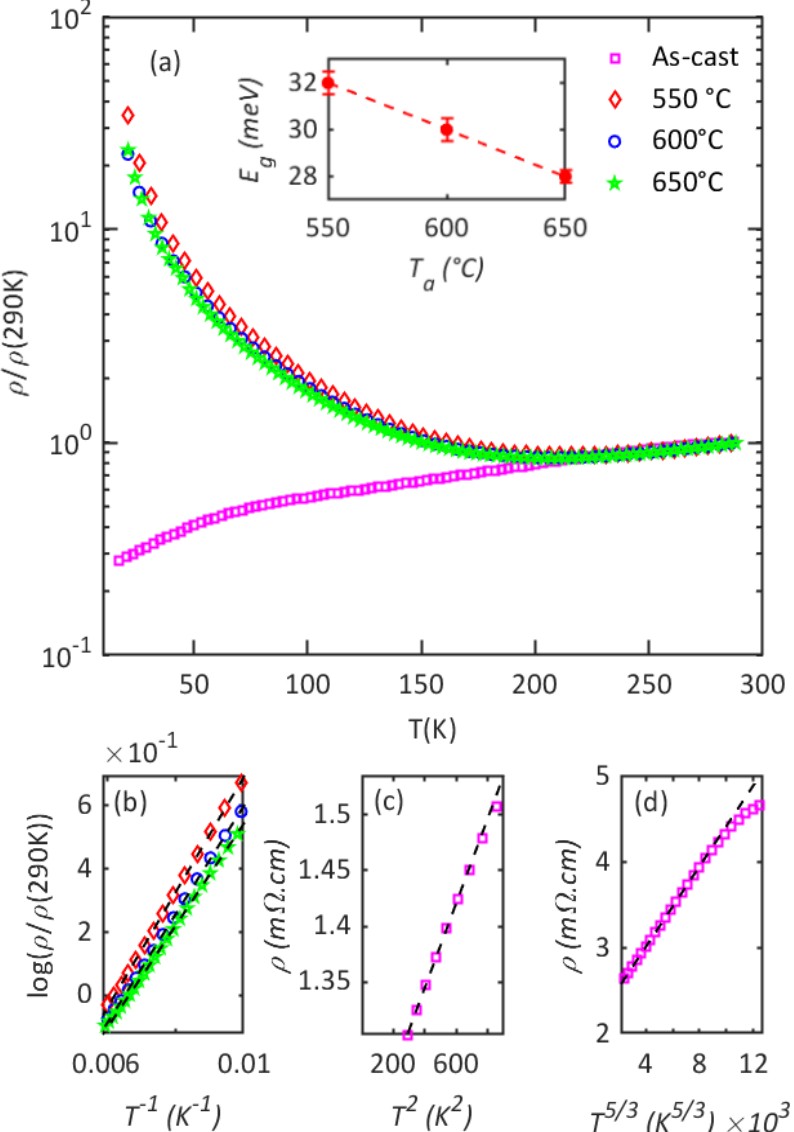

Figure 5: Figure (a): Temperature-dependence of normalized electrical resistivity of FeGa$_3$ annealed at different T$_a$. The inset shows the temperature-dependence of the energy gap from the in-gap donor state to the conduction band, lines are guide for the eyes. Figure (b): Arrhenius plot at 100 K to 160 K temperature range for the annealed samples. Figure (c): Low temperature T$^2$-dependence of electrical resistivity of as-cast sample. Figure (d): Comparison with T$^{5/3}$ dependence of electrical resistivity of as-cast sample.

tor intrinsic gap (see $E_g$ in figure 6). From our $\rho(T)$, we estimated the impurity activation gap ($E_g$) (plotted in the inset of figure 5 (a)). The decrease of $E_g$ with the annealing temperature before Fe$_{1+\delta}$Ga$_3$ amorphized suggests that the localized impurity states go towards the conduction band while its bandwidth is reduced, as result of the decrease of localized impurity concentration.

On the other hand, the high concentration of defects in the as-cast sample, the delocalization of the in-gap states, gives rise to a metallic behavior. The Fermi-liquid behavior found in

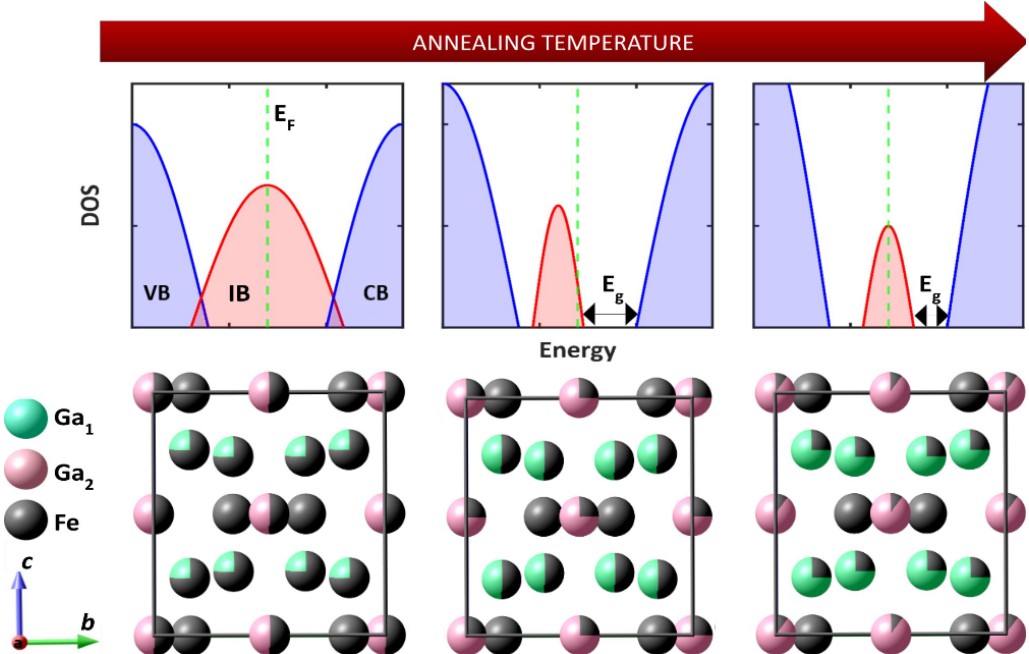

Figure 6: Schematic representation of electronic band (upper) and unit cell (lower) of $Fe_{1+\delta}Ga_3$ as function of annealing temperature. CB = conduction band, VB = valence band, IB = intermediate in-gap band and $E_g$ is the extrinsic gap, that decreases with the annealing temperature. The semi-filled spheres represents the amount of Fe antisite disorder at the Ga positions.

$\rho(T)$ as-cast sample reminisces of a problem involving defect-induce dirty metal, for which the additional free carriers induced by disorder contributes to the increase of paramagnetic susceptibility and enhancement of the momentum relaxation [27]. However, the large value of the quadratic coefficient in the FL-fitting (see fig. 5 (c)) $A_f \approx 0.38(2)\mu\Omega cm/K^2$ can not exclude the scenario of a strong renormalization in the effective mass likely induced by the presence of strong electronic correlations or spin fluctuations. In addition the $T^{5/3}$ dependence of the $\rho$ in a wide range of temperatures, within 90-230 K (see Fig. 5 (d)) might indicate followed by slow saturation, indicates the presence of spin fluctuations, which could also cause the renormalization of the quadratic coefficient at low temperatures. Spin fluctuations are also observed in the metallic-ferromagnetic phase of $FeGa_{3-x}Ge_x$ to high dopant concentration and mediate marginal Fermi-liquid state on the border of ferromagnetism near the critical point [19]. The presence of spin fluctuations induced by strong antisite disorder in our as-cast samples recalls the scenario that spin dynamics might play a relevant role in the quantum critical behavior and non-canonical quantum critical material phase of $FeGa_{3-x}Ge_x$ system, which is tuned to a putative FM-QPC by random substitution of Ge [19]. Our findings trigger future low-temperature investigation of the magnetic properties of $Fe_{1+\delta}Ga_3$ to reveal the nature of the magnetic ground state competing with spin fluctuations mechanism in a system with antisite disorder features.

## 5 Conclusion

The effect of controlled antisite disorder in $FeGa_3$ on the electronic, magnetic and crystallographic structure has been studied in polycrystalline $Fe_{1+\delta}Ga_3$ samples with a slight excess of Fe identified as Fe antisite disorder.

Fe-defects induce paramagnetic and metallic states at ambient temperature. Tiny decrease of impurities induced by annealing reduces the magnetic susceptibility and changes the ground state from a disordered correlated metal to a magnetic narrow-gap semiconductor. The later is possibly due to the transition of a delocalized impurity band to localized in-gap states as the number of defects approaches the percolation threshold. These findings contributes to a better understanding of the role of disorder in FeGa$_3$ and motivates further studies of the nature of the disordered-driven semiconductor-metal transition in FeGa$_3$ and the role of spin-fluctuation and strong correlations.

# Acknowledgements

We acknowledge M.C.A. Fantini for the access to the LCr-IFUSP.

**Funding information**    JLJ acknowledges support of JP-FAPESP (2018/08845-3) and CNPq-PQ2 (310065/2021-6), C.K.R acknowledges the support of FAPESP (2019/24522-2), VM acknowledges the support of JP-FAPESP (2018/19420-3).

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
