# Peer review of "Investigation of role of antisite disorder in pristine cage compound FeGa3"

_SciPost Physics Proceedings, doi:SciPost Phys. Proc. 11, 023 (2023)_

## Round 1 · Referee Report · Anonymous (Referee 1) · 2023-1-1

Strengths

  • Complete study of the effect of annealing on electronic properties FeGa3.
  • Well written.

Weaknesses

1: The interpretation of the measured resistivity of the as-grown sample is not very solid.

Report

The paper discusses work presented at the SCES conference and has not been published before.

Requested changes

1: in the last paragraph on page 3, the structure factor S(q) is discussed. It is not clear what the difference is between pristine poly-crystals and results presented. It is also not clear how the data for the pristine sample is obtained. Is this from a different study? More details would be useful. The last sentence of the same paragraph is also not very clearly formulated. Please rewrite.

2: On page 4 the SON is discussed. It is not clear from the description how it is obtained from the XRD data. A reference would be useful.

3: On page 5 it is stated that defects induce a magnetic state. What is this based on? All magnetization data is paramagnetic. The susceptibility increases, but that does not imply a magnetic state.

4: On page 5, the sentence “This metal-semiconductor transition oppositely…” is unclear. Please reformulate.

5: On page 5 it is stated that the ‘large Af indicates a strong renormalization of the effective electronic mass’. This conclusion is unlikely. It is much more likely that defects result in additional free carriers that (i) contribute to increased paramagnetic susceptibility and enhanced momentum relaxation. This can be explained within textbook framework and doesn’t require reference to electronic correlations or spin fluctuations. The latter is speculation. It would be better to put possible alternative interpretations in the discussion section only.

6: Figure 5 requires labels (a,b, etc.). The caption of the figure refers to ‘not-annealed’, please replace with as-cast for consistency.

7: page 7, metal-insulator -> metal-semiconductor.

---

## Round 2 · Author Response

List of changes
1 - We have included a detailed description of the calculation of the structure factor S(q).
2- We have included a detailed description on how SON are obtained from XRD Rietveld analysis
3- We change the claiming that defects induce a magnetic state, substituting by paramagnetic state and including a more detailed discussion of the magnetic properties of our compound.
4- We have included a careful discussion on the electrical resistivity of the as-cast grown sample, including a new explanation to the large Af.
5- We correct the labels in the figures 5.

---

## Round 2 · List of Changes

1 - We have included a detailed description of the calculation of the structure factor S(q).
2- We have included a detailed description on how SON are obtained from XRD Rietveld analysis
3- We change the claiming that defects induce a magnetic state, substituting by paramagnetic state and including a more detailed discussion of the magnetic properties of our compound.
4- We have included a careful discussion on the electrical resistivity of the as-cast grown sample, including a new explanation to the large Af.
5- We correct the labels in the figures 5.

---

## Editorial Decision

published